# Agricultural Land Quality Evaluation and Utilization Zoning Based on the Production–Ecology–Health Dimension: A Case Study of Huanghua City

**Fan Wang [1,2], Pengtao Zhang [3], Guijun Zhang [2,3,*] and Jiahao Cui [3]**

[1]  School of Resources and Environmental Science, Hebei Agricultural University, Baoding 071001, China; 15127613872@163.com
[2]  Key Laboratory of Agriculture and Environmental Protection in Hebei Province, Baoding 071001, China
[3]  School of Land Resources, Hebei Agricultural University, Baoding 071001, China; zhangpt@hebau.edu.cn (P.Z.); cuijiahao036@163.com (J.C.)
*  Correspondence: zhgj@hebau.edu.cn

**Abstract:** Clarifying the constituent elements of agricultural land quality, carrying out multi-dimensional quality evaluation of agricultural land, and implementing precise land consolidation and utilization zoning all have important guiding significance for achieving efficient utilization of agricultural land in China. This work analyzed the multi-dimensional evaluation framework of agricultural land based on its comprehensive quality elements and the production, ecological, and health functions. This paper constructed a multi-dimensional agricultural land "production–ecology–health" quality evaluation index system and evaluation criteria, and carried out a multi-dimensional quality evaluation of agricultural land in Huanghua City, Hebei Province, China. The spatial superposition of each dimension's evaluation results, combined with the logical relationship between agricultural land use and each dimension's quality, realized the renovation and utilization zoning of agricultural land. The results are as follows: (1) The production, ecological, and health qualities of agricultural land in Huanghua City were below the average and there is spatial variability, whose proportions of grade III and below were 63.12, 66.23, and 69.32%, respectively. In addition, low score areas are mainly located in the south and northwest of the study area. (2) The obstacle factors to quality in different dimensions were different: The obstacle factors to production quality were matter content, soil pH, irrigation guarantee rate, and alkaline hydrolysis nitrogen; groundwater salinity and depth, soil pH, and chemical fertilizers consumption for ecological quality; and groundwater salinity and depth and soil pH for health quality. (3) Agricultural land in Huanghua city is divided into five types of remediation, including 30,277.34 hm² for high efficiency utilization area, 10,576.54 hm² for production quality cultivation area, 34,387.86 hm² for health quality cultivation area, and 56,311.22 hm² for comprehensive consolidation and restoration area; special remediation measures are proposed for different types of zones. The work improves the multi-objective quality evaluation index system for agricultural land and implements differentiated land remediation strategies by identifying obstacle factors through zoning. It provides methodological ideas to improve the efficiency of land remediation and utilization.

**Keywords:** agricultural land quality; production–ecology–health; multi-dimensional evaluation; utilization zoning; Huanghua City



## 1. Introduction

As the basic means of production and resources, agricultural land is important to national food security and economic development [1]. However, agricultural land has been occupied and the comprehensive quality has declined with the advancement of industrialization and urbanization in China. Therefore, the contradiction between economic

development and agricultural land use has become prominent. China's relevant departments issued a series of regulations and standards in the 21st century: Regulations for Gradation on Agriculture Land Quality (GB/T 28407-2012), Regulations for Agricultural Land Grading (GB/T 28405-2012), Soil Environmental Quality Standard' (GB 15618-2008), and Technical Specification for Farmland Productivity Investigation and Quality Evaluation (NY/T1634-2008). The "trinity" monitoring and evaluation management system of agricultural land quantity, quality, and ecology provides a basis for its safe utilization and monitoring protection. In addition, the report of the 20th National Congress of the Communist Party of China pointed out that promoting the rural revitalization strategy, consolidating the foundation of food security omnidirectionally, and striving for high-quality development are necessary. Therefore, multi-dimensional and accurate evaluation of the comprehensive quality of agricultural land, the implementation of zoning, and consolidation and management measures are the basis and necessary conditions for the rural revitalization strategy.

With the continuous changes in the meaning and extension of agricultural land quality, the definition and evaluation system of agricultural land quality have gradually shifted from single to comprehensive, going through three stages according to times requirements: natural quality evaluation, natural economic quality evaluation, and multiple quality evaluation [2]. Different stakeholders have varying understandings of the meaning of agricultural land quality. These studies either focus on the production quality of agricultural land from the natural conditions, utilization conditions, and investment levels of agricultural land [3–5], or focus on the ecological quality of agricultural land from the soil environment and the invasion of harmful substances [6,7]. Few studies have focused on the capacity of agricultural land to provide services based on the other needs of service recipients, such as focusing on the health quality of the products provided by agricultural land. Therefore, this study believes that the understanding of agricultural land quality should start from the needs of human beings at all levels, and comprehensively define the level of agricultural land's ability to provide various products and services to humans. Furthermore, from the perspective of human demand for agricultural land function, this study defines the meaning of agricultural land quality as: In the process of using agricultural land for direct or indirect agricultural production, the degree or ability of agricultural land can directly or indirectly provide material production function, ecological protection function, and health service function. Early studies focused on the comprehensive evaluation of soil production quality and land ecological environment, including climate [8], soil physico-chemical properties [9], soil microorganisms [10], irrigation water [11], fertilizer use [12], and land use [13,14]. Research in the 21st century aims to cultivate healthy soil and improve soil productivity. The following index systems constructed by various countries provide alternative index sets for farmland soil health assessment and management in different regions [15,16]: Cornell's Comprehensive Assessment of Soil Health [17], New Zealand's Soil Indicators (SINDI) method [18], and the Muencheberg Soil Quality Rating (M-SQR) method developed by the Leibniz Agricultural Landscape Research Center, Germany [19].

Research on agricultural land quality evaluation in China focuses on the following aspects: (1) Cultivated land quality assessment. The single- or multi-objective evaluation was carried out on the soil production quality [20], ecological quality [21], environmental quality [22], and management quality [23]. (2) Cultivated land productivity was calculated to manage cultivated land balance of occupancy and replenishment. Wei et al. proposed to strengthen the quality assessment of supplementary cultivated land based on productivity [24]. In addition, land quality evaluation methods and models are diversified, including the comprehensive index method [25], 3D magic model [26], BP neural network model [27], and SVM model [28]. Based on the definition of agricultural land quality from the perspective of human demand for agricultural land services, this study constructed a theoretical framework for agricultural land quality assessment, which was established from the perspectives of ensuring food security, improving production conditions, maintaining

balance of nature, protecting soil and water resources, sustainable use, and providing health products.

Agricultural land focuses on land remediation zoning with zoning research, which designs consolidation zoning plans based on the natural, ecological, and socio-economic conditions and quality improvement potential of the land [29,30]. In addition, different attributes and using management zoning are divided from landscape patterns [31], noise pollution [32], policy effectiveness [33], utilization efficiency [34], and farming management [35]. However, most studies neglect the different requirements of each dimension quality on the same factor, and the influence of the spatial correlation of each quality factor on the implementation of scale regulation and management. Therefore, in this study, the strictest criterion of the same factor in the quality requirements of each dimension is taken as the common evaluation criterion. Combined with the evaluation results, taking the scale renovation and management as the goal, the study took advantage of spatial superposition method, according to the spatial consistency of obstacle factors, accurate positioning, and rational zoning. This study aims to propose a multi-dimensional comprehensive quality evaluation framework and method for agricultural land, identify the obstacles and their limitations in different dimensions, and provide a basis for accurate zoning of agricultural land remediation and utilization. We propose to construct a comprehensive quality evaluation framework for agricultural land from the three dimensions of "production–ecology–health", and construct an evaluation index system in different dimensions. In order to verify the application effect of the method system on the quality management of agricultural land, we took Huanghua City of China as an example, fully considering the diverse needs of the recipients of agricultural production services, and based on the production quality and ecological quality that have been studied, taking into account the multi-dimensional needs of human beings for agricultural land, implemented a comprehensive quality evaluation based on the three dimensions of "production–ecology–health". This paper identifies the obstacles and limitations in different quality dimensions, and based on the evaluation results and the consistency of various quality factors in land space, implements a study on the zoning of agricultural land remediation. It can provide practical reference for the implementation of multi-objective quality monitoring and precise zoning management of agricultural land. Specifically, our detailed research objectives are: (1) to evaluate and grade the quality of agricultural land in the three dimensions of "production–ecology–health", and to clarify the differences in the spatial distribution of quality in each dimension; (2) to identify obstacle factors and degree of limitation in different quality dimensions; (3) to use the results of multi-dimensional evaluation to accurately implement agricultural land remediation and utilization zoning. Under the background of the increasingly prominent contradiction between economic development and the functional safety of agricultural land, this study has practical significance for the comprehensive quality safety of agricultural land and human food safety. It can also provide scientific basis for local governments to improve the safety quality of agricultural work, carry out agricultural land rectification, protection, and monitoring work, and help ensure the food security of the people.

## 2. Materials and Methods

### 2.1. Research Area

The work took Huanghua City (Figure 1), Hebei Province, China as the research area. The city is located in the southeast of Hebei Province, on the west coast of Bohai Bay. The urban area is $23.914 \times 10^4$ hm$^2$, and agricultural land accounted for 65.21% thereof. The geomorphic type is sea-retreating siltation and alluvial plain, with low-lying and flat terrain. It belongs to a warm temperate semi-humid monsoon climate with obvious marine climate characteristics, and the annual average temperature is 12 °C. The average annual rainfall is 656.5 mm, concentrated in July and August, with an average annual evaporation of 1980 mm. The difference between dry and wet soil is large, and the degree of salinity is high. The freshwater shortage is not conducive to agricultural production.

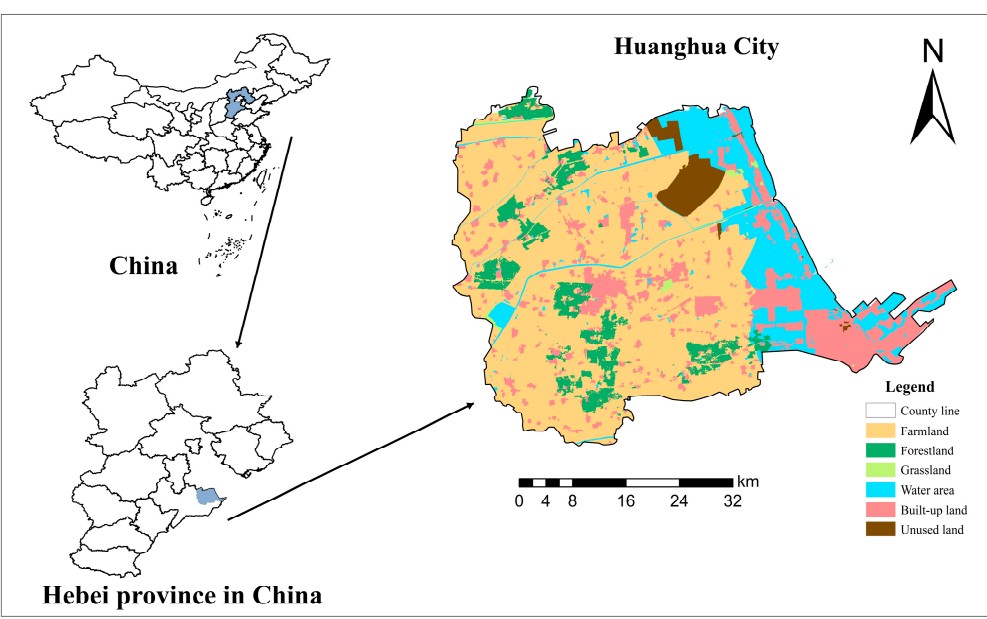

**Figure 1.** Location map of Huanghua City.

### 2.2. Data Sources

The adopted data are as follows: (1) Land use actuality data in 2020 with a resolution of 30 m downloaded from the Resource and Environmental Science Data Center of the Chinese Academy of Sciences (http://www.resdc.cn, accessed on 21 March 2021); (2) Data on pesticide and chemical fertilizer application rates obtained by field research in Huanghua City; (3) Agricultural land grading database in 2021; (4) Soil physical, chemical, and biological properties of 306 samples obtained by combining the results of the agricultural geological survey in Hebei Province in 2015 with the measured data of supplementary sampling in 2020–2021. The work took the land use status survey database of Huanghua City in 2020 as the base map, and the $30 \times 30$ m grid as the evaluation unit. The 1,601,182 evaluation units were determined after deducting water, construction, and unused lands, with a total area of 144,106.38 $hm^2$.

### 2.3. Research Ideas

Firstly, the work established a 3D (production–ecology–health) quality evaluation theoretical framework of agricultural land based on the characteristic of agricultural land quality components and their production, ecological, and health functions. Secondly, the evaluation index system of agricultural land quality was constructed from these dimensions, and the index grade was divided by the national standard and the existing research results. The fractal dimension evaluation was carried out by the weighted sum method. Then, the obstacle degree model was used to determine the factors and degrees that restricted agricultural land quality. Finally, the agricultural land renovation partitions were divided and differentiated maintenances were carried out according to the multidimensional quality evaluation results of agricultural land and the logical relationship between each dimension.

### 2.4. Evaluation Framework System of Agricultural Land Quality in Different Dimensions

With the development of social economy and the progress of ecological civilization, human demands for resources, environment, and food safety are constantly increasing. The excessive pursuit of high grain yield in traditional agricultural production cannot meet the needs of green agricultural development in the new era. Farmland management must simultaneously achieve coordinated development of agricultural production, environmental protection, and food safety. Therefore, based on the elements and multiple functions of agricultural land, combined with the characteristics of agricultural land in Huanghua City,

the study summarized the composition of water and soil resources, infrastructure, and utilization and management and other elements, as well as their impact process and direction on the production, ecology, and health functions of agricultural land (Figure 2), providing a theoretical basis for the construction of the "production–ecology–health" quality evaluation system.

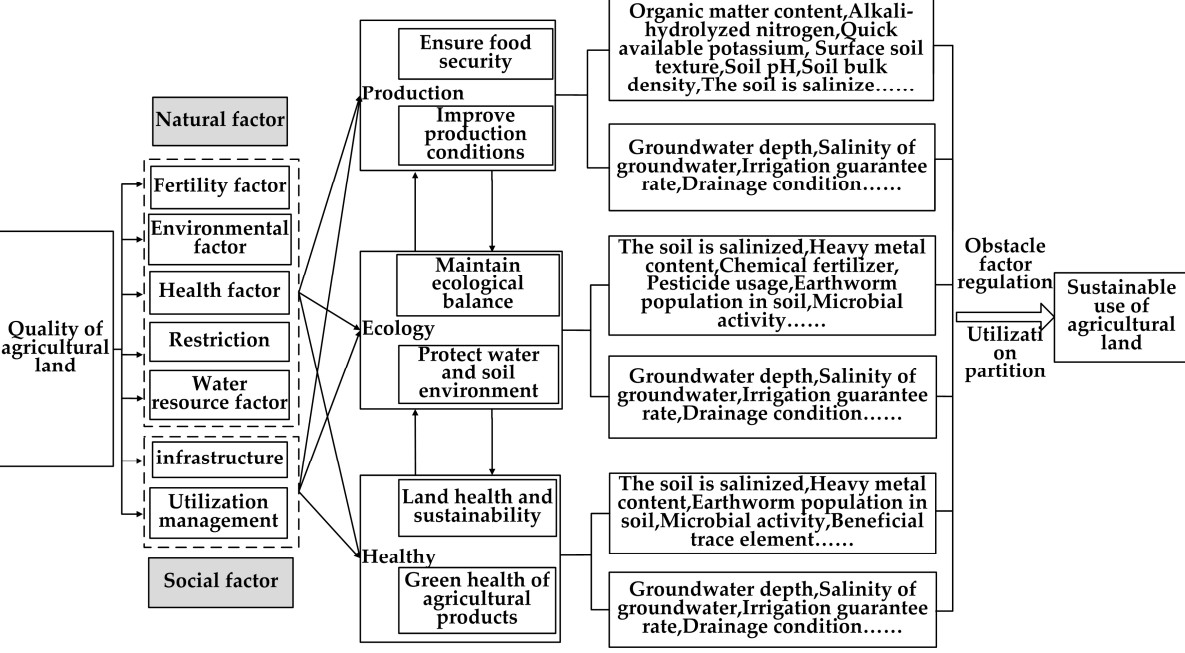

**Figure 2.** Production–ecology–health quality evaluation framework for agricultural land.

As a core element to characterize the quality and production capacity of agricultural land, soil fertility, environment, and health can improve biological productivity, environmental quality, and animal and plant health. Water resources can ensure the normal growth of crops, while their environment and utilization can react to the physical and chemical properties of soils. Infrastructure is the fundamental condition for the normal operation of agricultural production activities, which can improve production. Human management of farmland affects agricultural land's sustainability. Farmers optimize the limiting factors of agricultural production through technology to increase the efficient and safe use of agricultural land and improve crop yields. The functional requirements of human beings for agricultural land have shifted from ensuring food production to safety and health to the development of a social economy and people's living standards.

The production quality of agricultural land is the foundation and core of the farmland system, which determines the input and output of agricultural production and the guarantee to meet the food demands. Eco-quality is the ecological service function of agricultural land as an ecological subsystem. As the premise of agricultural land's sustainable use, it reflects the stability of the farmland ecosystem [36]. Health quality refers to the maintenance of the organizational structure, autonomy, and resilience of the farmland system [37], which affects the sustainable use of agricultural land resources and the quality and safety of agricultural products.

## 2.5. Evaluation Index System Construction

The production quality of agricultural land reflects the level of agricultural productivity and considers soil fertility, soil environment, water supply, and infrastructure facilities that affect crop yield. Eco-quality reflects the ability and state of agricultural land to maintain the stability of the ecosystem, considering the soil environment, ecosystem stability, water resources environment, infrastructure allocation, and utilization management level. Health quality focuses on maintaining production capacity and improving the farmland

environment and plant and animal health [38]. It considers the soil and water resources environment, functional health, and system stability, and beneficial trace elements of selenium, iodine, and fluorine are included in the evaluation index system (Table 1).

**Table 1.** Comprehensive production–ecology–health quality evaluation index system of agricultural land in Huanghua City.

| Criterion Layer | Index | Production Quality | Index Weight | Eco-Quality | Index Weight | Health Quality | Index Weight |
|---|---|---|---|---|---|---|---|
| Soil fertility | Organic content | √ | 0.12 | √ | 0.06 | | |
| | Rapidly available potassium | √ | 0.06 | | | | |
| | Available Phosphorus | √ | 0.05 | | | | |
| | Alkaline hydrolysis nitrogen | √ | 0.10 | | | | |
| Soil environment | pH | √ | 0.08 | √ | 0.08 | √ | 0.08 |
| | Soil salinity | √ | 0.06 | √ | 0.10 | √ | 0.06 |
| | Volume weight of soil | √ | 0.07 | | | | |
| | Topsoil texture | √ | 0.05 | | | | |
| | Soil profile configuration | √ | 0.05 | | | | |
| Soil ecology | Soil microbial activity | | | √ | 0.09 | √ | 0.08 |
| | Soil earthworms' number | | | √ | 0.08 | √ | 0.07 |
| Soil health | Selenium | | | | | √ | 0.07 |
| | Iodine | | | | | √ | 0.06 |
| | Fluorine | | | | | √ | 0.06 |
| Water resource environment | Groundwater depth | √ | 0.07 | √ | 0.08 | √ | 0.07 |
| | Mineralized degree of groundwater | √ | 0.09 | √ | 0.09 | √ | 0.08 |
| Utilization management | Heavy metal pollution | | | √ | 0.10 | √ | 0.10 |
| | Fertilizer input | | | √ | 0.08 | √ | 0.07 |
| | Pesticide growth rate | | | √ | 0.05 | √ | 0.06 |
| Infrastructure | Draining conditions | √ | 0.09 | √ | 0.10 | √ | 0.08 |
| | Probability of irrigation | √ | 0.11 | √ | 0.09 | √ | 0.06 |

### 2.6. Index Evaluation Method

The work used the inverse distance weighted interpolation (IDW) of ArcGIS 10.7 software's statistical module to draw the spatial distribution map of sample data. The grading evaluation rules of the participating indices determine every index weight: the comprehensive score of agricultural land quality in each dimension was calculated by the scoring method, while the grades were divided into grades I to V (grade I is optimal) by the equal interval method.

(1)    Index classification assignment

Index classification assignment rules referred to national regulations, industry standards, and related research results, e.g., Land Quality Geochemical Assessment Regulations (DZ/T0295-2016), Cultivated Land Quality Investigation and Monitoring Evaluation Regulation (Trial Draft), and Gradation Regulation on Agriculture Land Quality (GBT 28407-2012). The Index Classification Rule Table (Table 2) was formulated according to the actual investigation test values of each index in Huanghua City, and the heavy metal pollution index was graded by the calculated heavy metal pollution index.

**Table 2.** Score assignment of production–ecology–health quality evaluation index of agricultural land in Huanghua City.

| Criterion Layer | Index | Index Classification and Assignment Criteria | | | | | | | | | |
|---|---|---|---|---|---|---|---|---|---|---|---|
| | | 100 | 90 | 80 | 70 | 60 | 50 | 40 | 30 | 10 | 0 |
| Soil fertility | Organic matter content (g/kg) | >40 | >30–40 | >20–30 | >10–20 | >6–10 | ≤6 | | | | |
| | Rapidly available potassium (mg/kg) | >200 | >150–200 | >100–150 | >50–100 | >30–50 | ≤30 | | | | |
| | Available phosphorus (mg/kg) | >40 | >20–40 | >10–20 | >5–10 | >3–5 | ≤3 | | | | |
| | Available nitrogen (mg/kg) | >150 | >120–150 | >90–120 | >60–90 | ≤60 | | | | | |
| Soil environment | pH | 6.0–7.9 | 5.5–6.0 or 7.9–8.5 | 5.0–5.5 or 8.5–9.0 | | 4.5–5.0 | | | <4.5 or 9.0–9.5 | ≥9.5 | |
| | Soil salinity (g/kg) | <1 | ≥1–<2 | | ≥2–<4 | | | ≥4–<6 | | ≥6 | |
| | Volume weight of soil (g/cm³) | 1–1.25 | <1 or 1.25–1.35 | | 1.35–1.45 | | 1.45–1.55 | | >1.55 | | |
| | Topsoil texture | Soil | | Clay | Sand | | | Chisley soil | | | |
| | Soil profile configuration | Soil Soil/sand/soil | Soil/clay/soil | | Sand/clay/sand Soil/clay/clay Soil/sand/sand | Sand/clay/clay | Clay/sand/clay Clay Clay/sand/sand | Sand Gravel | | | |
| Soil ecology | Soil microbial activity | >Mean of sample points | | | Mean of sample points | | | | <Mean of sample points | | |
| | Number of soil earthworms (bar/m³) | >Mean of sample points | | | Mean of sample points | | | | <Mean of sample points | | |
| Soil health | Se (g/kg) | 0.4–3.0 | | 0.175–0.40 | | | | 0.125–0.175 | ≤0.125>3.0 | | |
| | Iodine (g/kg) | 5–100 | | 1.50–5 | | | | 1–1.50 | ≤ 1 or >100 | | |
| | Fluorine (g/kg) | 550–700 | | 500–550 | | | | 400–500 | ≤400 or >700 | | |
| Water resource environment | Groundwater depth (m) | 4.0–5.0 | 3.0–4.0 or 5.0–6.40 | | 2.50–3.0 | | 1.8–2.5 | | ≤1.0 or >6.4 | | |
| | Salinity of groundwater (g/L) | <1 | 1–3 | | 3–10 | | 10–50 | | >50 | | |
| Utilization management | Heavy metal pollution | $F_j \le 1$ | | | $1 < F_j \le 2$ | | $2 < F_j \le 3$ | | $3 < F_j \le 5$ | | $F_j \ge 5$ |
| | Fertilizer input (kg/hm²) | | | | 225 | | | | | | |
| | Pesticide growth rate (%) | >−2 | −1–2 | | 0–1 | | 0–1 | >1 | | | |

**Table 2.** *Cont.*

| Criterion Layer | Index | Index Classification and Assignment Criteria | | | | | | | | | |
|---|---|---|---|---|---|---|---|---|---|---|---|
| | | 100 | 90 | 80 | 70 | 60 | 50 | 40 | 30 | 10 | 0 |
| Infrastructure | Drainage condition | The drainage system is sound and there are no floods. | The drainage system is basically sound, and the water accumulates for <2 days after heavy rains in the wet year. | | The drainage system is general, and the water accumulates for ≥2–3 days after heavy rains in the wet year. | | | Without the drainage system, the water accumulates for ≥3 days after heavy rains in the general year. | | | |
| | Probability of irrigation (%) | ≥90 | ≤70–90 | | ≤30–70 | | | ≤30 | | | |

Note: $F_j$ is the Nemerow pollution index.

The heavy-metal pollution indices of As, Hg, Cu, Zn, Ni, Pb, Cd, and Cr in agricultural land were calculated by the Nemerow pollution index method. The calculation model is:

$$F_i = \frac{C_i}{B_i} \tag{1}$$

$$F_j = \sqrt{(F_{i,ave}^2 + F_{i,max}^2)/2} \tag{2}$$

where $F_i$ is the single-factor pollution index; $C_i$ is the measured value of the heavy metals; $B_i$ is the evaluation standard of heavy metals; $F_j$ is the Nemerow pollution index; $F_{i,ave}$ is the average value of the single-factor pollution index; and $F_{i,max}$ is the maximum value of the single-factor pollution index.

(2)    Calculation of the quality score for each dimensional Criterion layer

The quality scores for each dimensional criterion layer were calculated by the weighted summation method. The formula is as follows:

$$F_i = \sum_{j=1}^{m} W_{ij} S_{ij} (j = 1, 2, 3 \ldots) \tag{3}$$

where $F_i$ is the criterion layer quality score for each dimension quality; $W_{ij}$ is the weight of each index; and $S_{ij}$ is the score of each index.

(3)    Comprehensive score calculation of each dimension quality

The comprehensive scores of agricultural land quality in each dimension were calculated by the weighted summation method. The formula is as follows:

$$A_i = \sum_{j=1}^{n} W_{ij} S_{ij} (i = 1, 2, 3 \ldots) \tag{4}$$

where $A_i$ is the comprehensive score of each dimension quality; $W_{ij}$ is the weight of each index; and $S_{ij}$ is the score of each index.

The work takes the comprehensive weight of the analytic hierarchy process and entropy weight method as the final weight of each evaluation index to improve the rationality of the calculation. The calculation formula is as follows:

$$W_{ij} = \frac{E_{i1} \cdot E_{i2}}{\sum_{i=1}^{n} E_{i1} \cdot E_{i2}} \tag{5}$$

where $W_{ij}$ is index weight; $E_{i1}$ is weight calculated by the analytic hierarchy process; and $E_{i2}$ is weight calculated by the entropy weight method.

(4)    Quality grading methods

The study ranked the overall quality scores of the different quality dimensions of each module from smallest to largest, and classified them into levels I–V according to the Equal Interval Method.

### 2.7. Obstacle Factor Diagnosis Methods

The obstacle degree of each index is calculated by the obstacle degree model, which identifies the obstacle factors of agricultural land quality. The calculation model is:

$$D_{ij} = \frac{W_{ij} \cdot R_{ij}}{\sum_{i=1}^{n} W_{ij} \cdot R_{ij}} \times 100\% \tag{6}$$

where $W_{ij}$ is the contribution degree of the factor, i.e., index weight; $R_{ij}$ is the deviation degree of the index, which indicates the gap between the $i$ index and the optimal ideal value and is expressed by $1 - A_{ij}$; $A_{ij}$ is the standardized index value; and $D_{ij}$ is the obstacle degree of the index. According to the calculation results of obstacle degree, it was divided

into four types [39]: no restriction (0%), mild restriction (0–10%), moderate restriction (10–20%), and severe restriction (>20%).

*2.8. Agricultural Land Use Zoning Method*

The spatial and spatial differences in the constituent elements of agro-geological quality lead to regional differences in quality. Therefore, it is necessary to combine the evaluation results and logical relationships of the various dimensions (production, ecology, and health) of agricultural land with the spatial superposition method used for the zoning of land consolidation and utilization. High-efficiency utilization areas refer to better production, ecology, and health quality. Production quality cultivation areas have both ecological quality and health quality at level III or above and production quality below level III. Ecological quality cultivation areas have both production quality and health quality at level III or above and ecological quality below level III. Health quality cultivation areas have high production and ecological quality but poor health quality. Areas of comprehensive consolidation and restoration have at least one of the three below level III (Table 3).

**Table 3.** Zoning rules for agricultural land use in Huanghua City.

| Zoning Type of Agricultural Land Use | Production Quality | Eco-Quality | Health Quality |
|---|---|---|---|
| Efficient utilization area | I, II, and III | I, II, and III | I, II, and III |
| Production quality cultivation area | - | I, II, and III | I, II, and III |
| Ecological quality cultivation area | I, II, and III | - | I, II, and III |
| Health quality cultivation area | I, II, and III | I, II, and III | - |
| Comprehensive remediation cultivation area | IV and V | IV and V | IV and V |

## 3. Results

*3.1. Comprehensive Quality Evaluation Results of Production, Ecology, and Health of Agricultural Land*

A multi-dimensional evaluation of agricultural land in Huanghua City was performed based on the above methods in the work. The production–ecology–health quality was divided into levels I–V, and the obstacle measurement results were analyzed separately (Tables 4 and 5 and Figure 3).

The production quality score of agricultural land in Huanghua City is between 64 and 83.3. The lowest score for the soil environment is 20.53 and the highest score is 31.4, which is the highest mean score and the most varied, with spatially interspersed areas of low and high scores; the lowest score is for the water resource environment, with an average score of only 8.52, showing a trend of gradual improvement from coastal to inland. Based on grading results according to the Equal Interval Method, it can be seen that the level-II horizontal area is the largest and accounts for 25% of the total agricultural land area. They are mainly distributed in the eastern part of Huanghua City, such as Nandagang Management Area, Zhongjie Friendship Farm, and Jiucheng Town. Levels III–V account for 22%, 21%, and 20% of the total agricultural land area, respectively, and are distributed in the western townships of Huanghua City. The Level-I horizontal area is the smallest and accounts for only 12%. It has a small number and is sporadically distributed in the Nandagang Management Area, Zhongjie Friendship Farm, and Qijiawu Town. Factors with an obstacle degree > 10% were selected as the main obstacle factor to study the production quality of agricultural land in Huanghua City. Therefore, organic matter content, soil pH, irrigation guarantee rate, and alkaline nitrogen decomposition are the main obstacle factors, with obstacle degrees of 15.58%, 13.94%, 12.69%, and 12.04%.

**Table 4.** Statistical table of the evaluation results of each dimensional criterion layer for agricultural land in Huanghua City.

| Dimensional Criterion | Production Quality | | | Ecological Quality | | | Health Quality | | |
|---|---|---|---|---|---|---|---|---|---|
| | Minimum Score | Maximum Score | Minimum Score | Maximum Score | Minimum Score | Maximum Score | Minimum Score | Maximum Score | Mean Score |
| Soil fertility | 21.42 | 27.61 | 23.97 | 3.65 | 4.82 | 4.15 | / | / | / |
| Soil environment | 20.53 | 31.40 | 27.67 | 8.25 | 18.23 | 14.03 | 7.80 | 14.12 | 11.32 |
| Soil ecology | / | / | / | 5.14 | 17.15 | 12.33 | 4.52 | 15.08 | 10.88 |
| Soil health | / | / | / | / | / | / | 10.37 | 56.83 | 34.13 |
| Water resource environment | 6.27 | 14.42 | 8.52 | 6.79 | 15.31 | 9.15 | 5.93 | 13.52 | 8.06 |
| Utilization management | / | / | / | 11.71 | 19.28 | 15.31 | 8.94 | 14.67 | 11.63 |
| Infrastructure | 10.70 | 18.91 | 13.31 | 10.60 | 18.11 | 13.25 | 7.37 | 12.43 | 9.15 |

**Table 5.** Production–ecology–health quality evaluation results of agricultural land in Huanghua City.

| Level | Production Quality | | | Ecological Quality | | | Health Quality | | |
|---|---|---|---|---|---|---|---|---|---|
| | Number of Units | Area (hm²) | Proportion (%) | Number of Units | Area (hm²) | Proportion (%) | Number of Units | Area (hm²) | Proportion (%) |
| Level I | 185,761 | 16,718.49 | 12% | 165,136 | 14,862.24 | 10% | 200,978 | 18,088.02 | 13% |
| Level II | 404,763 | 36,428.67 | 25% | 375,580 | 33,802.20 | 23% | 290,251 | 26,122.59 | 18% |
| Level III | 347,033 | 31,232.97 | 22% | 481,550 | 43,339.50 | 30% | 266,098 | 23,948.82 | 17% |
| Level IV | 336,569 | 30,291.21 | 21% | 393,948 | 35,455.32 | 25% | 373,968 | 33,657.12 | 23% |
| Level V | 327,056 | 29,435.04 | 20% | 184,968 | 16,647.12 | 12% | 469,887 | 42,289.83 | 29% |

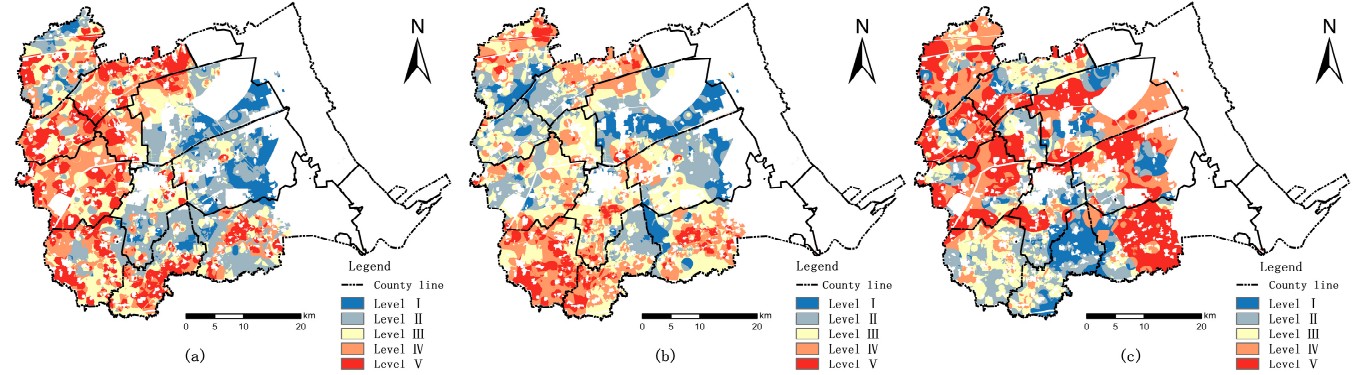

**Figure 3.** Production–ecology–health quality evaluation results of agricultural land in Huanghua City: (**a**) production quality; (**b**) ecological quality; (**c**) health quality.

The ecological quality score of agricultural land is between 55 and 85.6. The lowest score is for soil fertility, with an average score of only 4.15. The relatively low score areas are located in Huanghua Town, Changguo Town, and Old Town. The criterion layer with the greatest difference in scores is soil ecology, with scores ranging from 5.14 to 17.15. It can be seen that the distribution of soil ecology on agricultural land in Huanghua is extremely uneven. The high-score areas are mainly concentrated in the western, central, and eastern regions of Huanghua City, while the low-score area is mainly concentrated in the northern and southern regions. The level-III horizontal area is the largest and accounts for 30%; areas of levels I and V are low and account for only 10 and 12%, respectively. The area proportions occupied by levels II and IV are similar with 23% and 25%, respectively. Factors with an obstacle degree >10% were still selected as the main obstacle factors. The main obstacles to ecological quality are groundwater mineralization, groundwater burial depth, soil pH, and fertilizer usage, with obstacle degrees of 15.94%, 15.54%, 15.12%, and 13.26%.

The health quality score of agricultural land is between 59 and 93.2. Among the guideline layers, the soil health score is the highest, ranging from 10.37 to 56.83, and it is also the most unevenly distributed layer, showing a distribution of low north–south

and high east–west; the lowest score is for infrastructure, with an average score of only 9.15, and the low score area is widely distributed in Qijiabu Township. The health quality of agricultural land has a large area of level V and level IV, accounting for 29% and 23%, respectively. They are mainly distributed in the northwest and southeast of Huanghua City. The level-I horizontal area is the smallest (accounting for 13%) and is mainly distributed in Jiucheng Town in the southern part of Huanghua City. The level-II and III horizontal areas have a low proportion and account for 18 and 17%, respectively. They are mainly distributed in Changguo Town in the south and Lvqiao Town in the north. The main barriers to health quality are groundwater mineralization, groundwater burial depth, and soil pH with obstacle degrees of 12.49%, 12.03%, and 11.49%.

### 3.2. Zoning for Agricultural Land Use

Agricultural land is divided into five parts by combining the evaluation results of various dimensions (production, ecology, and health) with the actual situation of Huanghua City. The zoning principle for agricultural land use is also an important standard (Table 3). It divides agricultural land into high-efficiency utilization areas, production quality cultivation areas, ecological quality cultivation areas, healthy quality cultivation areas, and comprehensive consolidation and restoration areas (Figure 4, Table 6).

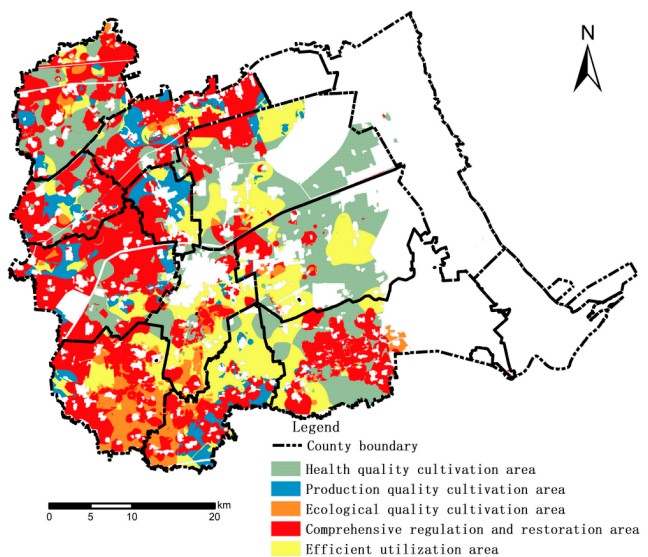

**Figure 4.** Zoning results of agricultural land consolidation and utilization in Huanghua City.

**Table 6.** Zoning types of agricultural land consolidation and restoration in Huanghua City.

| Partition Type | Area (hm²) | Proportion |
| --- | --- | --- |
| High-efficiency utilization area | 30,277.34 | 21.01% |
| Health quality cultivation area | 34,387.86 | 23.86% |
| Production quality cultivation area | 10,576.54 | 7.34% |
| Ecological quality cultivation area | 12,553.42 | 8.71% |
| Comprehensive Consolidation and Restoration area | 56,311.22 | 39.08% |
| Total | 144,106.38 | 100.00% |

The high-efficiency utilization area covers 30,277.34 km² and accounts for 21.01% of the total agricultural land area. Water and soil resource endowment in the district is high, with perfect basic supporting facilities and excellent comprehensive quality. This area is mainly distributed in Zhongjie Friendship Farm, Nandagang Management Area, and Jiucheng town. Therefore, the development of the area focuses on utilization and should give full play to its advantages. Production and income are increased by cultivating green organic agricultural products according to local conditions in high-efficiency utilization areas. In

addition, regular supervision and infrastructure maintenance should be strengthened with the existing agricultural geological quantity maintained.

The production quality cultivation area, which accounts for 7.34% of the total agricultural land area, covers 10,576.54 km$^2$. The area has a high ecological and health quality and is mainly distributed in the old towns, Yangerzhuang Town and Lvqiao Town. Agricultural production is seriously hindered due to low soil fertility in the zone, the insufficient irrigation guarantee rate, and high soil pH in some areas. Therefore, management activities should be carried out to improve the productivity of agricultural land. Efforts have been made to solve the obstacles affecting agricultural real estate through measures such as water-saving irrigation and increased application of organic fertilizers.

The ecological quality cultivation area is 12,553.42 km$^2$ and accounts for 8.71% of the total agricultural land. Groundwater mineralization degree in the zone is higher, and groundwater burial depth is shallower. This area is mainly concentrated in Changguo Town, Jiucheng Town, and Huanghua Town, where the excessive use of fertilizer has led to poor soil ecological environment. Therefore, measures for the consolidation and restoration of agricultural land in this area should focus on the ecological function. Reducing the burden on the soil requires a reduction and enhancement plan for fertilizers and pesticides. Groundwater mineralization and groundwater burial depth are closely related to soil salinization. Soil salinization impact can be reduced through open ditch drainage and concealed pipe salinization measures. This plan has the following advantages of increasing soil organic nutrients, protecting soil biodiversity, and ensuring the stability of agricultural land ecosystems.

The health quality cultivation area covers 34,387.86 hm$^2$ (accounting for 23.86%) and is mainly concentrated in Zhongjie Friendship Farm, Nandagang Management Area, Yangerzhuang Town, and Qijiawu Town. Soil pH in this area is high and the groundwater environment is poor. Healthy quality cultivation areas need to consider the interaction between the physical and chemical properties of agricultural land and external land use management. Sea ice water irrigation technology can be used to alleviate drought and block raised soil salinity. Use and cultivation are combined; the stalks are crushed and subjected to deep tillage to improve the self-purification, resistance, and resilience of agricultural land.

The comprehensive consolidation and restoration area is the zoning type with the largest proportion (39.08%). This area is mainly distributed in the western part of Huanghua City and the Yangerzhuang Town. The background conditions of agricultural land in the area are poor, with low utilization efficiency and comprehensive quality. Such areas should take improving agroecological conditions as their near-term goal and gradually increase production capacity after the ecological environment is optimized and stabilized. The ultimate goal is increasing production capacity and taking the production of healthy products as a constraint. In addition, secondary zoning can be carried out according to the differences in the main quality limiting factors. The consolidation and restoration measures should be refined in this area to improve accuracy. The area is subdivided into seven types of second-level zoning (Table 7) based on the first-level consolidation and restoration of partitions with differences in land consolidation and upgrading objects. It is conducive to zoning policies and improving land consolidation and restoration.

**Table 7.** Secondary zoning types of comprehensive agricultural land consolidation and restoration areas in Huanghua City.

| Secondary Partition Type | Area (hm$^2$) | Proportion |
|---|---|---|
| Key production quality improvement area | 126.78 | 0.23% |
| Key ecological quality improvement area | 214.35 | 0.38% |
| Key health quality improvement area | 89.51 | 0.16% |
| Key quality improvement area of production and ecology | 16,272.72 | 29.07% |
| Key quality improvement area of production and health | 16,586.55 | 29.63% |
| Key quality improvement area of ecology and health | 7162.45 | 12.80% |
| Key quality improvement area of production, ecology, and health | 15,858.87 | 28.33% |

## 4. Discussion

(1) There is a wealth of research on comprehensive land quality evaluation. The index system constructed covers soil physical, chemical, and biological indicators, but most of them only focus on the land function. Indicators of production functions are evaluated. This type of research only focuses on the relationship between the quality level of cultivated land and crop yield; the biological characteristics of the soil in the evaluation indicators have not attracted enough attention. This study attempts to improve the evaluation system by adding two indicators: soil microbial activity and soil earthworm quantity. Nowadays, ecological safety and green development are the primary needs of development. Therefore, the establishment of a multi-objective and coordinated comprehensive land quality evaluation system has become a hot topic at the forefront of sustainable land use.

This work focused on the production, ecology, and health functions of agricultural land and considered multi-level goals such as improving farmland productivity, protecting farmland's ecological environment, and providing health products and services. An evaluation index system was constructed for a comprehensive quality evaluation. Finally, consolidation and restoration farmlands were zoned according to the evaluation results, which is conducive to the precise protection and management of farmland. Health dimension evaluation focused on the healthy production indicators in existing studies as well as the indicators of providing healthy quality products.

(2) Different regions have different factors that restrict the comprehensive quality of agricultural land. The research area in the work is in the coastal low plain, where agricultural production is significantly affected by soil salinization and groundwater mineralization. Therefore, the selection of evaluation indicators paid more attention to this aspect. The research mainly selected indicators such as soil pH, soil salinity, groundwater depth, and groundwater mineralization. The results of quality assessment and obstacle factor diagnosis show that soil pH has a significant impact on the quality of all dimensions, and is the main factor limiting the comprehensive quality of agricultural land in Huanghua City. The mineralization degree and buried depth of groundwater are significant factors that affect the ecological quality and health quality of agricultural land in Huanghua City. Practical research in other regions can increase or decrease indicators based on the actual agricultural geological conditions and agricultural production conditions of the selected area, and select an evaluation index system that is suitable for the local area.

(3) Insufficient research is due to limited data access. The work only selects three ecological quality and health quality evaluation indicators (fertilizer input, the growth rate of pesticides, and heavy metal pollution) to reflect the impact of utilization management on environmental quality in the work. Therefore, the practice of ecological quality and health quality evaluation should consider the possible environmental impact of agricultural and non-agricultural production activities in the research area.

## 5. Conclusions

(1) The production–ecology–health quality of agricultural land in Huanghua City was different and had spatial differentiation. The spatial distribution of the high score areas of production quality and ecological quality had certain coupling, mainly distributed in the eastern and northwestern regions of Huanghua City, while the low score areas thereof were distributed in the northwestern and southwestern regions of Huanghua City, respectively. The high score areas of health quality were mainly located in the southern part of Huanghua City, and the low score areas were mainly located in the northwest and southeast regions. There were many high score areas of health quality in the southern part of Huanghua City, while the low score areas were in the northwest and southeast.

(2) The obstacles that affected agricultural land quality in Huanghua City in all dimensions were mainly soil fertility, soil environment, water resource environment, utilization management, and infrastructure. Therefore, the practice of remediation and restoration should be based on the spatial differences of the main obstacle factors in order to implement precise zoning strategies. Production quality cultivation areas should pay attention

to implement water-saving irrigation, increasing manure and other measures; ecological quality cultivation areas should mainly focus on reducing soil salinization and heavy metal pollution; and health quality cultivation zone should pay attention to the combination of use and maintenance. For comprehensive remediation and restoration areas with large areas, there are many factors that hinder land quality. Secondary zoning can be carried out based on the differences in land remediation and improvement goals, and remediation and restoration planning can be carried out based on the main contradictions to improve the efficiency of land remediation and restoration.

This work implemented a quantitative evaluation based on the production–ecology–health dimension of agricultural land, identified obstacles and their spatial location of each dimension, and carried out rectification zoning and implemented targeted rectification measures based on the spatial relationships of the obstacles. It can provide experience for multi-objective agricultural geological quality monitoring and management in other regions as well as a reference for precise consolidation and restoration zoning.

**Author Contributions:** Conceptualization, F.W. and P.Z.; Data curation, G.Z.; Formal analysis, F.W.; Funding acquisition, G.Z.; Investigation, F.W. and J.C.; Methodology, F.W. and G.Z.; Project administration, G.Z.; Resources, G.Z.; Supervision, P.Z.; Validation, G.Z.; Visualization, F.W. and G.Z.; Writing—original draft, F.W.; Writing—review and editing, P.Z. and J.C. All authors have read and agreed to the published version of the manuscript.

**Funding:** This research was funded by the Social Science Foundation of Hebei Province, China (Grant No. HB19YJ020) and the Social Science Development Research Project of Hebei Province, China (Grant No. 20220202230).

**Data Availability Statement:** Not applicable.

**Acknowledgments:** We would like to express our gratitude and respect to the editors and anonymous reviewers for their invaluable comments and constructive suggestions that helped us improve the quality of the manuscript.

**Conflicts of Interest:** The authors declare no conflict of interest.

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
