# Peer review of "Agricultural Land Quality Evaluation and Utilization Zoning Based on the Production–Ecology–Health Dimension: A Case Study of Huanghua City"

_land, doi:10.3390/land12071367_

Round 1

Reviewer 1 Report

Wang et al. proposed a multi-dimensional quality evaluation framework for agricultural land. They also identified the obstacles and zoning patterns for Huanghua City. This work is important to understand and guide agricultural land management. And the results can be potentially useful for policymakers in the context of food security under climate change. I think it is publishable if several issues can be addressed. 

(1) The authors highlighted that most of land quality evaluations only focus on the land function, lacking attention to the biological characteristics of the soil as evaluation indicators. I wonder if the authors can do some comparisons between their methods here and other research in the discussion.

(2) It will be helpful to readers to know where Huanghua City is and why chose it as a case study by adding a larger-scale map to show the city's location and land use land cover information

(3) There are some sentences that are not easy to read and understand, for example, the first sentence in the abstract “The clear quality components of agricultural land production, ecology, and health as well as the development of multi-dimensional quality evaluation and differentiated consolidation and utilization zoning can guide the efficient use of agricultural land in China.” I think some sentences can be improved by making them short and clear. 

There are some sentences that are not easy to read and understand, for example, the first sentence in the abstract “The clear quality components of agricultural land production, ecology, and health as well as the development of multi-dimensional quality evaluation and differentiated consolidation and utilization zoning can guide the efficient use of agricultural land in China.” I think some sentences can be improved by making them short and clear. 

Author Response

Dear Reviewer,

    Hello, thank you very much for your valuable feedback on this article. We have made modifications based on your suggestions and provided modification instructions (see attachment).

    Thank you again for your valuable feedback.

    Sincerely yours,

    Guijun Zhang

Reviewer 2 Report

The authors explore the multi-objective quality evaluation and consolidation and restoration zoning plan of agricultural land and policy implication for the efficient utilization and differentiated consolidation of agricultural lands based on multi-source spatial data and different models. The findings are also very interesting and clearly presented. However, I have some reservations.

1. The abstract needs to refine the research conclusions according to the research results and further refine the agricultural land quality and its dimension.

2. The paper fails to state the meaning of agricultural land quality and the research gaps in existing literature and clearly point out its novelty and contributions. The authors mention "Cultivated land quality assessment, cultivated land productivity and evaluation methods and models". The statement is too general. It is hard to find theoretical or methodological contributions for this paper, as well as what are the new findings that this paper contribute to existing literature.

3. Figure 1. definitely requires further development. Please supply the basin basis. Why is “production-ecology-health” to characterize quality evaluation framework for agricultural land? The figure is not self-evident enough and the font format is not well.

4. Table 2. Score assignment of production-ecology-health quality evaluation index of agricultural land that is mainly highlighting soil health? The Criterion layer evaluation can further clarify.

5. In the discussion part, the influencing mechanism and utilization zoning of land quality evaluation should be connected with the front part and the relationship with the previous model analysis.

6. The conclusion part needs to be condensed, and it is the result of the paper. Conclusion part requires further development especially according to the obstacles that affected agricultural land quality or recommendations. The results of different models are substantial, and the significance of solving specific problems needs to be further strengthened.

English language is required further.

Author Response

Dear Reviewer,

    Hello, thank you very much for your valuable feedback on this article. We have made modifications based on your suggestions and provided modification instructions (see attachment).

    Thank you again for your valuable feedback.

    Sincerely yours,

    Guijun, Zhang

Reviewer 3 Report

Dear Authors,

Congratulations for your entire work and especially for the way you presented your findings in the submitted manuscript.

In my opinion, it can be published as it is - very good English and graphical presentation (figures and tables), very well written and explained formulas, good accent on Methodology and Results and so on.

Reading the entire paper, I noticed just a few minor things, such as:

1. In the text you mention ”Fig. ....”, while they are entitled ”Figure ...”.

2. In Table 2, please repeat the header row on the second page, otherwise the readers won't understant it. In the same table, what is and where does appear "Fj", mentioned as note at the end.

3. Table 5 isn't mentioned in the text and its title is on a different page as the table.

Except this aspects, I consider that the manuscript has a good structure and it can be published in its current form.

Good luck in your future research activity!

Author Response

Dear Reviewer,

    Hello, thank you very much for your valuable feedback on this article. We have made modifications based on your suggestions and provided modification instructions (see attachment).

    Thank you again for your valuable feedback

    Sincerely yours,

    Guijun Zhang

Round 2

Reviewer 2 Report

After modification, there has been some improvement, but the questions raised have not been fully answered. The new findings that this paper contribute to existing literature, and the answer still needs to be considered.

Author Response

Dear Reviewer,

    Thank you very much for your valuable feedback. We have made the modifications according to your suggestions (see attachment).

Sincerely yours,

Guijun Zhang
